# Healthcare Costs and Resource Utilisation of Italian Metastatic Non-Small Cell Lung Cancer Patients

**DOI:** 10.3390/cancers16030592

**Published:** 2024-01-30

**Authors:** Nicola Gentili, William Balzi, Flavia Foca, Valentina Danesi, Mattia Altini, Angelo Delmonte, Giuseppe Bronte, Lucio Crinò, Nicoletta De Luigi, Marita Mariotti, Alberto Verlicchi, Marco Angelo Burgio, Andrea Roncadori, Thomas Burke, Ilaria Massa

**Affiliations:** 1Outcome Research, Healthcare Administration, IRCCS Istituto Romagnolo per lo Studio dei Tumori (IRST) “Dino Amadori”, 47014 Meldola, Italy; nicola.gentili@irst.emr.it (N.G.); valentina.danesi@irst.emr.it (V.D.); andrea.roncadori@irst.emr.it (A.R.); ilaria.massa@irst.emr.it (I.M.); 2Unit of Biostatistics and Clinical Trials, IRCCS Istituto Romagnolo per lo Studio dei Tumori (IRST) “Dino Amadori”, 47014 Meldola, Italy; flavia.foca@irst.emr.it; 3Healthcare Administration, Azienda Unità Sanitaria Locale della Romagna, 48121 Ravenna, Italy; mattia.altini@regione.emilia-romagna.it; 4Department of Medical Oncology, IRCCS Istituto Romagnolo per lo Studio dei Tumori (IRST) “Dino Amadori”, 47014 Meldola, Italy; angelo.delmonte@irst.emr.it (A.D.); g.bronte@staff.univpm.it (G.B.); lucio.crino@irst.emr.it (L.C.); marita.mariotti@irst.emr.it (M.M.); alberto.verlicchi@irst.emr.it (A.V.); marco.burgio@irst.emr.it (M.A.B.); 5Ospedale di Stato della Repubblica di San Marino, 47893 San Marino City, San Marino; nicoletta.deluigi@iss.sm; 6MSD Innovation & Development GmbH, 8004 Zurich, Switzerland

**Keywords:** non-small cell lung cancer (NSCLC), immunotherapy, real-world evidence, healthcare resource utilisation (HCRU), costs, economics, administrative databases, Italy

## Abstract

**Simple Summary:**

This study aimed to assess the costs and healthcare resource utilization (HCRU) of metastatic non-small cell lung cancer (NSCLC) by biomarker status in the pre- and post-approval of the immuno-oncology agent. The analysis examined healthcare costs and HCRU before and after the local regulatory approval of pembrolizumab as a first-line (1L) treatment. Patients were stratified into mutation-positive and negative/unknown groups according to mutational status. The negative/unknown group was further sub-grouped based on the availability of pembrolizumab at the time of starting 1L treatment. Costs and HCRU were analyzed separately for the 1L treatment and overall disease follow-up across lines of therapy and by groups. The study found that introducing 1L immunotherapy has improved overall survival, but healthcare spending has increased concurrently. Decision-makers may find our results useful in deciding how best to allocate resources for treating metastatic NSCLC in terms of the health–economic model and policy.

**Abstract:**

This study evaluated the economic burden of metastatic non-small cell lung cancer patients before and after the availability of an immuno-oncology (IO) regimen as a first-line (1L) treatment. Patients from 2014 to 2020 were categorized according to mutational status into mutation-positive and negative/unknown groups, which were further divided into pre-1L IO and post-1L IO sub-groups depending on the availability of pembrolizumab monotherapy in 1L. Healthcare costs and HCRU for a 1L treatment and overall follow-up were reported as the mean total and per-month cost per patient by groups. Of 644 patients, 125were mutation-positive and 519 negative/unknown (229 and 290 in pre- and post-1L IO, respectively). The mean total per-patient cost in 1L was lower in pre- (EUR 7804) and post-1L IO (EUR 19,301) than the mutation-positive group (EUR 45,247), persisting throughout overall disease follow-up. However, this difference was less when analyzing monthly costs. Therapy costs were the primary driver in 1L, while hospitalization costs rose during follow-up. In both mutation-positive and post-IO 1L groups, the 1L costs represented a significant portion (70.1% and 66.3%, respectively) of the total costs in the overall follow-up. Pembrolizumab introduction increased expenses but improved survival. Higher hospitalisation and emergency room occupation rates during follow-up reflected worsening clinical conditions of the negative/unknown group than the mutation-positive population.

## 1. Introduction

Globally, lung cancer is the most frequent tumour and the leading cause of cancer-related deaths [1]. In Italy, over 43,900 patients receive a lung cancer diagnosis each year [2]. In particular, non-small cell lung cancer (NSCLC) accounts for approximately 80–85% of all lung cancer diagnoses, with a five-year survival rate of less than 5% for Stage IV disease [3,4]. The high incidence rate of severe disability associated with NSCLC disease is a significant public health issue. Furthermore, the high level of healthcare resource utilisation (HCRU) incurred by NSCLC patients and the high costs of new pharmacological treatment can result in financial pressure for healthcare payers [5,6]. Therefore, the estimate of healthcare costs of NSCLC patients is of great interest nowadays for sound decision-making in the allocation of limited resources to provide the best care in an economically sustainable context. Cost analyses on the management of NSCLC have been examined more and more worldwide, mainly in the USA. However, most papers were limited in assessing the economic value of specific drugs focusing on subgroups of patients with a targetable genetic aberration, or on specific histological subtypes (i.e., squamous or squamous), rather than examining the overall NSCLC category or broader treatment patterns [7,8,9,10,11,12]. Above all, most studies have been conducted before the approval of newer costly therapies such as immunotherapies, which have changed the overall medical management of metastatic NCSLC in recent years. This new class of medicine has improved clinical outcomes, but healthcare spending has increased concurrently [13,14,15,16,17,18]. Unfortunately, estimations of the financial burden associated with the management of NSCLC, especially after the approval of new costly therapies such as immune checkpoint inhibitors (ICIs), are lacking in Italy.

Perrone et al. performed one of the first Italian analyses of NSCLC costs in 2004, but the reliability of their findings is now outdated [19]. Another Italian study reported an increase in the economic burden of metastatic NSCLC patients as opposed to previous analyses [20]. A real-world analysis evaluated the management of NSCLC cases diagnosed in the Veneto region in 2015 and 2017 [21]. The study detected a 38% rise in the average overall cost for the 2017 cohort compared to the 2015 cohort due to the introduction of new expensive oncologic drugs for the care of metastatic diseases Likewise, an economic assessment of advanced NSCLC patients treated with either pembrolizumab or tyrosine-kinase inhibitors (TKI) estimated average per-patient healthcare costs of EUR 51,735 and EUR 30,708 during the first year of the first-line (1L) treatment, respectively [22]. A recent cost comparison across Europe showed that the mean per-patient cost related to Italian advanced NSCLC patients who received two or more lines of therapy amounted to EUR 19,317 [23].

Following market access approval by the Italian Medicines Agency, pembrolizumab was authorized in the Emilia–Romagna region in July 2017 as a 1L treatment for metastatic NSCLC patients with a programmed death-ligand 1 (PD-L1) Tumour Proportion Score (TPS) of at least 50% and no targetable mutations. The introduction of immuno-oncology (IO) treatment in the 1L setting has significantly changed the overall medical and economic management of metastatic NSCLC patients. Considering this recent therapeutic innovation, this study evaluated medical costs and the HCRU of metastatic NSCLC before and after the approval of a 1L immunotherapy agent in the Emilia–Romagna region (July 2017).

## 2. Materials and Methods

### 2.1. Study Design

This evaluation is an economic addendum analysis of a retro-prospective observational study conducted in IRCCS Istituto Romagnolo per lo studio dei Tumori (IRST) “Dino Amadori” (located in the Emilia–Romagna region), which investigated the clinical outcomes of metastatic NSCLC patients before and after the regional regulatory approval of PD-L1 inhibitors in the 1L setting of NSCLC [24]. In this economic assessment, costs and HCRU were evaluated in the same population reported by Danesi et al., adding the patients with oncogenic driver mutations who were excluded in the previous manuscript [24]. The perspective of the National Healthcare Service (SSN, Servizio Sanitario Nazionale) was adopted while only considering costs sustained by the healthcare payer. Costs and HCRU were expressed as per patient per month (PPPM) and mean total per-patient cost. The analysis was conducted separately for the 1L treatment and overall disease follow-up (FU) across lines of therapy. The first-line period was calculated as the time from the start of the 1L drug administration until the first of the following event: 30 days after the conclusion of the 1L treatment, the start of the second-line therapy, the end of the observational period (December 2020), the death, or the last visit. The overall disease FU period was calculated as the time from the beginning of the 1L drug administration and the end of the observational period (December 2020), the death, or the last visit. The study evaluated costs associated with all–cause hospitalisations, cancer therapies, and outpatient and hospice care. Data on HCRU included the number of ordinary hospitalisations, the relative length of stay (LOS), the number of drug administrations, and outpatient and Emergency Room (ER) admissions. In addition, the LOS of hospice admission was assessed.

### 2.2. Study Population

Patients included in this economic analysis had the same eligibility criteria as in the previous study [24]: (i) aged ≥18 years (ii) with a confirmed diagnosis of NSCLC presenting with Stage IV or Stage IIIB with a rapidly progressive disease (IIIBrp). These patients also experienced disease progression to Stage IV within six months from the first anticancer treatment without completing both radiotherapy and chemotherapy induction therapy. (iii) Residents of the Emilia–Romagna region (iv) initiated 1L treatment between 1 January 2014 and 30 June 2020 at IRST, and (v) patients were enrolled after signing the informed consent or after death. In comparison to the previous clinical study [24], we excluded patients who died within 30 days of starting 1L treatment to avoid overestimation of costs and HCRU, as well as patients who were enrolled in clinical trials during the study period, because costs were reimbursed by study sponsors. The study subjects were recruited in two phases: before (pre-1L IO) and after (post-1L IO) the approval of pembrolizumab as a 1L treatment in the Emilia–Romagna region in July 2017. The observational period for each patient started at the diagnosis time of Stage IV/IIIBrp and ended with the patient’s death or the end of the observational period (December 2020).

### 2.3. Cohorts Description

The overall population was stratified by mutation status of epidermal growth factor receptor (EGFR) or anaplastic lymphoma kinase (ALK) or receptor tyrosine kinase (ROS1) (EGFR/ALK/ROS1 mutation-positive vs. EGFR/ALK/ROS1 negative or unknown). The patients without oncogenic driver mutations or whose status were unknown were further sub-grouped according to the availability of the first ICI available as 1L monotherapy in the Emilia–Romagna region (pembrolizumab in PD-L1 TPS > 50% metastatic NSCLC) at the date of initiation of 1L systemic anticancer treatment (pre- and post-1L IO, respectively). In short, the medical cost and HCRU were calculated separately for the following patient groups:

Mutation-positive with oncogenic driver mutation in EGFR, ALK, or ROS1, regardless of the availability of the first ICI as 1L monotherapy at the date of their 1L starting time.Negative/Unknown without oncogenic driver mutation or unknown status in EGFR, ALK, or ROS1, which was divided into two sub-groups according to the availability of the first ICI as 1L monotherapy:Pre-1L IO included eligible patients who started 1L treatment from January 2014 to June 2017 before the 1L ICI was available in the Emilia–Romagna region;Post-1L IO included eligible patients who started 1L treatment from July 2017 to June 2020 after 1L ICI was available in the Emilia–Romagna region.

### 2.4. Data Sources

Patients diagnosed with metastatic NSCLC between January 2014 and June 2020 were recruited from the IRST Electronic Health Record (EHR). The clinical dataset was obtained from data registered by physicians in EHR during routine clinical practice. EHR contains visits, routine laboratory examinations, disease assessments, administered drugs, and all the procedures that NSCLC patients could receive in outpatient and inpatient settings. Data on hospitalisation, drug prescriptions, outpatients and ER visits, and hospice care were obtained from multiple administrative databases:

Hospital Discharge Records (SDO) collects information on hospital admissions, both ordinary (with at least one overnight stay in hospital) and day-hospital stays (admissions without an overnight stay), which was active until April 2016. The SDO collects the start and end date of hospitalisation, the primary diagnosis coded according to the International Classification of Disease, Ninth Revision, Clinical Modification (ICD-9-CM), and the procedures and services provided. The remuneration system is based on the classification of the Diagnosis Related Groups (DRG), which aggregates the activities of each individual diagnosis and defines the reimbursement rate. Under the DRG-based reimbursement system, each hospitalized patient falls into a group of homogeneous diagnostic cases. Therefore, patients with the same DRG value have been allocated the same reimbursement costs, which do not correspond to the total amount of resources used during the hospital stay, but it is an average value of resource utilisation attributable to that DRG [25];Outpatient Specialist Assistance Database (ASA) collects individual information on all outpatient visits, clinical tests, and procedures delivered in the outpatient setting. The outpatient costs were estimated based on the assumption that each procedure is reimbursed according to the Regional Healthcare Range of Fees [26]. The ASA costs were calculated by multiplying the unit cost for resource consumption;Emergency Room Admissions Database (PS) contained information about any single emergency admission, including procedures, diagnoses, and costs performed during emergency room (ER) admission;Electronic Health Records were used to retrieve data on biomarker and gene panel tests.Pharmaceutical Databases (FED and AFT–direct hospital administration and territorial pharmacies distribution) coded according to the Anatomical Therapeutic Chemical classification system were used to collect data on drugs administered;Hospice Discharge Records contain the main information about any single hospice admission;Registry of Mortality (REM) of the Emilia–Romagna region was used to retrieve data on vital status.

The assignment of a patient identification code to all Emilia–Romagna inhabitants, regardless of admission setting (inpatient or outpatient), is an enabled deterministic record linkage among these various databases.

### 2.5. Outcome Measures

Costs and HCRU associated with the 1L therapy and the overall disease FU were estimated separately for the three groups of patients. A mean per-patient total and monthly costs were estimated for the following categories:

Ordinary hospitalization refers to costs of all-cause hospitalization (with at least one overnight stay in hospital), except inpatient stays for therapy administration (identified by code 410);Cancer therapy included costs of dispensed drugs, ordinary hospitalization, and day-hospital service for therapy administration (Code 410), and the costs associated with the outpatient setting of drug administration (Code 99.25), medical visits, and blood draws performed before each drug administration;Outpatient procedures included costs associated with FU visits, diagnostic exams, biomarker and gene panel tests, laboratory tests, and day-hospital admissions (with code different from 410) performed in the outpatient setting;Hospice included all costs associated with the hospice admission.

For HCRU, we collected the number and LOS of ordinary all-cause hospitalizations except for therapy administration (Code 410), the number of drug administrations (ordinary and day-hospital admission with Code 410, and outpatient visits with Code 99.25), the number of outpatient visits (except access, which reported Code 99.25), the LOS of hospice care, and the number of all-cause ER admissions.

### 2.6. Statistical Analysis

Categorical data were presented as frequencies and percentages, while continuous data were summarized using median and minimum–maximum values for patient demographic and treatment characteristics. Mean PPPM was reported for HCRU and costs, considering the period of 1L treatment and the overall disease FU across the lines of treatments. Mean total cost and HCRU per patient were also reported, as well as the percentile distribution to better observe the data distribution. All the analyses were conducted using SAS 9.4 software (SAS Institute, Cary, NC, USA).

## 3. Results

### 3.1. Patient Characteristics

A total of 644 patients were considered according to the study inclusion/exclusion criteria. We identified 125 patients with oncogenic driver mutations (EGFR, ALK, or ROS1 mutation-positive group) and 519 without mutations or unknown status (EGFR, ALK, or ROS1 negative/unknown group). The negative/unknown population was split into pre-1L IO (*N* = 229) and post-1L IO (*N* = 290) sub-groups. Patient demographic and clinical characteristics are summarized for each group in Table 1.

Most patients were over 70, with a median age of 70.6 (min–max: 35.7–89.9). In both pre- and post-1L IO groups, the majority of patients were males (≥65.1%). An opposite pattern was observed in the mutation-positive group, where females were 69.9%. Most patients in the negative/unknown group were smokers, with a similar proportion in pre- (94.8%) and post-1L IO (92.6%). Conversely, the number of “never” (50.9%) and “ever” (49.1%) smokers was similar in the mutation-positive group. A slight difference in ECOG (Eastern Cooperative Oncology Group) Performance Status (PS) at IIIBrp/IV stage diagnosis was observed between the mutation-positive and negative/unknown groups. The predominant histology was adenocarcinoma, in particular, in the mutation-positive group (96.0%), with a comparable proportion in pre- (75.6%) and post-1L IO (78.2%). Among the known metastatic sites, the contralateral lung was the most prevalent metastasis location in all three cohorts, accounting for more than one-third of patients.

### 3.2. Treatment Patterns

All 644 patients who met the study’s criteria received a 1L treatment. Due to the poor prognosis, only 37.4% and 11.2% of patients received second- and third-line treatment, respectively. Only 3.7% of patients received additional treatment beyond the third-line treatment (Appendix A).

In the mutation-positive cohort, 103 patients (82.4%) were treated with 1L targeted therapy according to their mutation status, while 17 (13.6%) and five (4.0%) patients received multi- and single-agent chemotherapy, respectively (Table 2).

In the negative/unknown cohort, multi-agent was the most common 1L regimen in pre- (67.7%) and post-1L IO (49.0%). In pre-1L IO, 74 patients (32.3%) were treated with single-agent therapy. In comparison, in post-1L IO, the 1L treatment most utilized after multi-agent chemotherapy was the PD-1/PD-L1 inhibitor single-agent pembrolizumab administrated to 67 patients (23.1%). Only 19 patients (6.5%) received pembrolizumab in combination with chemotherapy. The median duration of the 1L treatment was 10.1 months (min–max: 1.0–67.0) for the mutation-positive group, and 2.7 (min–max: 1.0–49.7) and and 3.8 (min–max: 1.0–41.2) months for the negative/unknown cohort in pre- and post-1L IO, respectively. The median duration of overall FU across lines of therapy was 17.5 months (min–max: 1.6–76.8) for the mutation-positive group, and 5.8 (min–max: 1.0–78.9) and 8.3 (min–max: 1.0–41.3) months for the negative/unknown cohort in pre- and post-1L IO, respectively.

### 3.3. Healthcare Costs

The mean total per-patient cost associated with 1L treatment in the mutation-positive group amounted to EUR 45,247, which resulted in a mean per-patient per-month cost (PPPM) of EUR 3814 (Table 3).

For the negative/unknown group, the mean total per-patient cost was lower, ranging from EUR 7804 in pre-IO 1L to EUR 19,301 in post-IO 1L, or EUR 3381 and EUR 3464 per-patient per-month cost, respectively (Table 3). The main cost driver was associated with cancer therapy, accounting for about 76.5% (EUR 34,597) and 64.9% (EUR 12,517) in the mutation-positive and post-1L IO groups, respectively. The higher costs associated with patients treated in the pre-1L IO group were mainly driven by cancer therapy (EUR 2790) and hospitalization (EUR 2789), representing 35.8% and 35.7% of the overall cost, respectively. The hospitalization expenditure associated with the negative/unknown group was remarkably high, especially in pre-1L IO. Regarding the outpatient procedures, the mean total per-patient cost ranged from a maximum of EUR 5967 in the mutation-positive group to a minimum of EUR 1988 in the pre-IO 1L group. However, when the PPPM spending was analyzed, the trend of the outpatient cost was the opposite, resulting in a monthly cost of EUR 573 for the mutation-positive group and EUR 576 and EUR 741 for pre- and post-IO 1L, respectively. The lowest costs were associated with hospice visits across all groups.

The mean total per-patient cost associated with the overall disease FU varied from EUR 70,985 in the mutation-positive group to EUR 19,649 and EUR 29,111 in pre- and post-IO 1L (Table 4).

It was remarkable that the expenditures associated with the 1L of treatment accounted for 63.7% and 66.3% of the total costs associated with the overall disease FU in the mutation-positive and post-IO 1L groups, respectively. Conversely, the mean per-patient total cost related to the 1L treatment of pre-IO 1L patients represented not even half (39.7%) of the total cost associated with the overall disease FU. Compared with the 1L treatment, even if the mean total per-patient cost of cancer therapy increased, a modest decrease in monthly expenditures was observed in drugs and outpatient visits across groups during overall disease follow-up (Table 3 and Table 4). Conversely, monthly costs related to hospitalization and hospice stays increased during the overall disease follow-up (Table 3 and Table 4).

In the pre-1L IO group, the monthly ordinary hospitalization cost (45.2%) exceeded that of drug therapy, becoming the primary cost driver during the overall disease follow-up (Table 4).

As in the 1L treatment, the cost associated with hospice continued to have a low budget impact on overall cost in all cohorts (Table 3 and Table 4).

### 3.4. HCRU

HCRU reported as the mean total number per person and PPPM showed low hospitalization in 1L and overall disease follow-up for all groups. However, both negative cohorts had fewer inpatient admissions than the mutation-positive group. The total LOS was similar across groups. The average length of hospitalization was between 6 and 8 days in 1L treatment (with a monthly hospitalization range of 0.1–0.3 days), and between 14 and 16 days during overall disease follow-up (monthly hospitalization range of 1.2–3.3 days) (Table 5 and Table 6).

The total number of cancer therapy administrations in the mutation-positive group varied from 17.4 in 1L treatment to 28.4 in overall disease follow-up (Table 5 and Table 6). Among negative/unknown patients, the total mean number of pharmacological treatments ranged from 16.2 to 13.6 in 1L and between 20.5 and 21.4 in overall disease follow-up (Table 5 and Table 6). The findings showed a marked difference in negative/unknown cohorts, where the mean PPPM number of cancer therapy administrations nearly doubled during 1L treatment in the pre-IO 1L group (4.8) compared to the post-IO 1L group (2.5). The same trend was observed throughout the entire disease follow-up, even if the gap narrowed (pre 2.94 vs. post 2.1).

Outpatient visits accounted for one of the greatest HCRU proportions in each group during 1L and the entire disease FU (Table 5 and Table 6). The mean number of total outpatient visits was much larger in the mutation-positive population than in the negative groups. However, focusing on HCRU per patient per month, the number of outpatient visits was similar across groups in 1L and entire disease FU (between 2.1 and 2.9).

We noticed hospice stay was short during 1L among all groups, increasing during overall disease FU, especially in pre-1L IO negative patients (Table 5 and Table 6).

The rate of ER visits was relatively low but higher in the negative/unknown pre-IO 1L group in overall disease follow-up (Table 5 and Table 6).

## 4. Discussion

The current study provided a real-world data analysis on the costs and HCRU of managing metastatic NSCLC in pre- and post-approval of IO therapy in the 1L setting. The analysis was conducted to investigate the costs and HCRU according to the mutational status of patients: patients with oncogenic driver mutations and patients without mutations or unknown status. Negative/unknown patients were further grouped in pre-1L IO and post-1L IO; that is, before and after the availability of the first ICI as 1L therapy in the Emilia–Romagna region (pembrolizumab in PD-L1 TPS ≥ 50%). Costs and HCRU were assessed as mean total per-patient and per-month costs separately for the 1L therapy and the overall disease follow-up across lines of therapy.

Although NSCLC disease is a significant public health issue regarding economic burden, little is known about trends in the cost of NSCLC management in Italy. To the best of our knowledge, this economic assessment of metastatic NSCLC disease is one of the few Italian studies conducted after the introduction of new ICIs, and that includes a resource-utilization analysis. Our study showed that the higher mean total per-patient costs were associated with the mutation-positive group, followed by post-1L IO and pre-1L IO in 1L treatment and overall disease FU. The differences in the clinical pathways and, more in general, in the approach to the disease among these three groups concern the use of different drugs, the duration of treatment, and the survival gain, which are the most responsible for this remarkable cost difference. Despite the high prices of new drugs, the cost difference was less pronounced among groups when PPPM costs were analyzed. This is because PPPM was not affected by treatment duration or prolonged survival as the mean total per-patient cost.

The findings showed that the growth in cancer drug prices was exceeded by more than half of the overall cancer spending. The high prices of these new cancer drugs influence current and future spending. However, these high prices may be legitimized if drugs may prolong survival and improve quality of life.

Most HCRU categories, such as hospital admissions for pharmacological treatment or outpatient services, decreased with the course of the disease. In contrast, other items, such as hospitalizations and hospice visits, grew over time.

It is not surprising that the costs sustained by the National Health Service for the treatment of metastatic NSCLC were mainly driven by oncological therapies, followed by hospitalizations and outpatient health services. This pattern is coherent with previous Italian investigations [20,22]. Due to patients’ worsening clinical conditions, hospitalization costs associated with the total disease FU become the most significant component of total costs in pre-1L IO. A large discrepancy in findings exists with the study of Migliorino et al., who reported a mean total PPPM halved from our 1L and total disease FU costs [20]. However, these disparities can be explained by the fact that these kinds of studies are sensitive to the period in which the studies are conducted, and the fact that the study of Migliorino was conducted in 2012, more than 10 years ago. Conversely, more similar to our findings are the results estimated by Buja et al. [21], even if their study population included only 52.07% of Stage IV NSCLC cases; it is also known that the cost associated with NSCLC disease increases as the disease progresses [27]. However, the reported total PPPM cost ranged from EUR 2601 for NSCLC patients diagnosed in 2015 to EUR 3611 for patients diagnosed in 2017 after adopting new drugs for metastatic patients [27]. Our findings on cancer therapy, in particular, for negative/unknown patients, are in line with costs estimated by Piantedosi et al., who reported a pharmaceutical expenditure per patient per month of EUR 1942 for mutation-positive populations and EUR 1316 for negative/unknown mutational status populations [28].

In line with the findings of a multinational investigation that included Italy [29], the most common setting for resource use was the outpatient context. Our study findings related to monthly HCRU showed that mutation-positive patients with oncogenic driver mutations (EGFR, ALK, or ROS1 mutations) seem to have a lower risk of hospitalization and ER visits than the negative/unknown population. However, further investigations are required.

The introduction of new agents in clinical practices increased oncological therapy costs associated with mutation-positive and post-1L IO cohorts. The mean per-patient total cost associated with post-1L IO is higher than those of pre-1L IO (+€ 11,497 in 1L). However, when we considered the mean per patient per month (PPPM), the findings were similar (+EUR 81 in 1L) due to a cost dilution for improving survival. Therefore, the availability of pembrolizumab in the 1L setting has increased the mean per-patient total cost, but it has demonstrated superior survival benefit, increasing the median overall survival from 6.2 months in pre-1L IO to 8.9 months in post-1L IO as documented by Danesi et al. [24]. This observation is consistent with the greater survival rate reported by previous studies on the efficacy of costly drugs included in updated NSCLC clinical pathways [30,31].

Our study has several limitations. First, the analysis collected data at the hospital level and did not capture potential costs for home care. This cost may be relevant, especially in the late phases of the disease, contributing to a healthcare cost increase. Moreover, indirect costs such as caregiver burden and lost workplace productivity are not evaluated. The strength of this study is the inclusion of an HCRU analysis based on detailed real-world data, which is scarce in previous Italian studies.

## 5. Conclusions

The current analysis provides real-world data on the cost and HCRU of NSCLC from the National Health Service perspective. The results demonstrate that the mutation-positive group has the highest costs, followed by the post-1L IO population and, lastly, the pre-1L IO population. This discrepancy in cost can be attributed to the introduction of new expensive anticancer treatments and the extension of survivability in mutation-positive and post-1L IO patients. The economic burden of mutation-positive and post-1L IO is extremely high during the first line of therapy, compared to the following treatment lines. However, when PPPM expenses are examined, the cost disparity across groups is less noticeable.

Our results confirmed that introducing 1L immunotherapy has improved overall survival, but healthcare spending has increased concurrently with anticancer treatments and hospitalization accounting for a considerable portion of total expenses. The follow-up data revealed that the negative/unknown group had worsening clinical conditions compared to the mutation-positive population, as evidenced by a higher rate of hospitalisation and ER visits.

This study is one of the first Italian analyses that covered detailed cost and HCRU data among metastatic NSCLC. In this context, real-world evidence is becoming increasingly significant, allowing for a better understanding of the cost-effectiveness of various therapeutic alternatives. Nonetheless, our findings are purely descriptive, and the data provided here may help in informing decision-makers to determine resource allocation in treating metastatic NSCLC regarding the health-economic model and policy.

## Figures and Tables

**Table 1 cancers-16-00592-t001:** Baseline demographic and clinical characteristics for mutation-positive patients with oncogenic driver mutations (EGFR, ALK, and ROS1 mutations) and patients without mutations (EGFR, ALK, and ROS1 negative or unknown), which were presented in two separate subgroups (pre- and post-1L IO).

	EGFR, ALK, or ROS1 Mutation-Positive Patients	EGFR, ALK, or ROS1 Negative/Unknown patients	
Characteristics	*N* = 125 (%)	Pre-1L IO *N* = 229 (%)	Post-1L IO *N* = 290 (%)
IIIBrp/IV stage			
IIIBrp	2 (2.6)	10 (4.4)	6 (2.1)
IV	123 (98.4)	219 (95.6)	284 (97.9)
Age at IIIBrp/IV stage diagnosis			
<70 years	69 (55.2)	118 (51.5)	124 (42.8)
70–74 years	18 (14.4)	52 (22.7)	67 (23.1)
75–79 years	13 (10.4)	35 (15.3)	70 (24.1)
80–84 years	18 (14.4)	22 (9.6)	20 (6.9)
≥85 years	7 (5.6)	2 (0.9)	9 (3.1)
Gender			
Female	87 (69.6)	80 (34.9)	93 (32.1)
Male	38 (30.4)	149 (65.1)	197 (67.9)
Race			
White	123 (98.4)	228 (99.6)	290 (100.0)
Other	2 (1.6)	1 (0.4)	0 (0.0)
Smoking history			
Never	57 (50.9)	9 (5.2)	17 (7.4)
Ever	55 (49.1)	165 (94.8)	212 (92.6)
Unknown	13	55	61
Year smoked			
≤20 years	7 (17.5)	9 (6.9)	17 (13.1)
>20 years	33 (82.5)	122 (93.1)	112 (86.9)
Unknown	40	131	129
Packs/year			
≤20 packs/years	14 (37.8)	11 (8.9)	20 (15.9)
>20 packs/years	23 (62.2)	113 (91.1)	106 (84.1)
Unknown	88	105	164
ECOG PS at IIIBrp/IV stage diagnosis			
0	26 (22.6)	37 (17.0)	42 (15.2)
1	67 (58.3)	143 (65.6)	186 (67.1)
≥2	22 (19.1)	38 (17.4)	49 (17.7)
Unknown	10	11	13
Histology			
Squamous cell	1 (0.8)	37 (16.4)	57 (20.0)
Non-squamous cell	121 (96.8)	172 (76.4)	223 (78.2)
*Adenocarcinoma*	*120 (96.0)*	*170 (75.6)*	*223 (78.2)*
*Large cell carcinoma*	*1 (0.8)*	*2 (0.8)*	*0 (0.0)*
Other	3 (2.4)	16 (7.2)	5 (1.8)
Unknown	0	4	5
Location of metastases			
Bone	41 (32.8)	80 (34.9)	79 (27.2)
Lymph nodes	37 (29.6)	49 (21.4)	78 (26.9)
Brain	30 (24.0)	35 (15.3)	49 (16.9)
Liver	14 (11.2)	23 (10.0)	26 (9.0)
Pleura	25 (20.0)	30 (13.1)	44 (15.2)
Contralateral lung	49 (39.2)	77 (33.6)	99 (34.1)
Other	15 (12.0)	67 (29.2)	45 (15.5)
Missing/Unknown	0	3	5

**Table 2 cancers-16-00592-t002:** First-line (1L) treatments administered by mutation status. The negative/unknown cohort was further grouped based on the 1L immune checkpoint inhibitor monotherapy availability at 1L starting time (pre-and post-1L IO).

	EGFR, ALK, or ROS1 Mutation-Positive Patients	EGFR, ALK, or ROS1 Negative/Unknown Patients	
First-line (1L) Therapies	*N* = 125 (%)	Pre-1L IO *N* = 229 (%)	Post-1L IO *N* = 290 (%)
Multi-agent chemotherapy	17 (13.6)	155 (67.7)	142 (49.0)
Gemcitabine + Platin	5 (4.0)	67 (29.3)	90 (31.0)
Pemetrexed +/− Platin	12 (9.6)	83 (36.2)	43 (14.9)
Paclitaxel + Carboplatin	—	5 (2.2)	9 (3.1)
Single-agent chemotherapy	5 (4.0)	74 (32.3)	62 (21.4)
Gemcitabine	4 (3.2)	42 (18.4)	30 (10.3)
Vinorelbine	1 (0.8)	28 (12.2)	31 (10.7)
Docetaxel	—	4 (1.7)	1 (0.4)
Targeted therapy	103 (82.4)	0 (0.0)	0 (0.0)
Afatinib	23 (18.4)	—	—
Alectinib	7 (5.6)	—	—
Crizotinib	7 (5.6)	—	—
Erlotinib	10 (8.0)	—	—
Gefitinib	37 (29.6)	—	—
Osimertinib	19 (15.2)	—	—
PD-1/PD-L1 inhibitor single agent	—	—	67 (23.1)
Pembrolizumab	—	—	67 (23.1)
PD-1/PDL1 inhibitor + chemotherapy	—	—	19 (6.5)

**Table 3 cancers-16-00592-t003:** Costs related to the 1L treatment are reported in euros (€) as mean per patient per month (PPPM) and other descriptive statistics, grouped by mutation status. The negative/unknown cohort was further sub-grouped based on 1L starting time in pre- and post-1L IO.

1L Costs	EGFR, ALK, or ROS1 Mutation-Positive Patients							EGFR, ALK, or ROS1 Negative/Unknown Patients Pre-1L IO							EGFR, ALK, or ROS1 Negative/Unknown Patients Post-1L IO						
	Mean PPPM € (%)	Mean Cost € (%)	Percentiles of the Costs Distribution					Mean PPPM € (%)	Mean Cost € (%)	Percentiles of the Costs Distribution					Mean PPPM € (%)	Mean Cost € (%)	Percentiles of the Costs Distribution				
			10%	25%	50%	75%	90%			10%	25%	50%	75%	90%			10%	25%	50%	75%	90%
Hospitalization	575 (15.1)	4415 (9.7)	0	0	1471	7500	13,297	1233 (37.2)	2789 (35.7)	0	0	0	4161	8405	1154 (33.3)	3181 (16.5)	0	0	0	4332	8799
Cancer therapy	2583 (67.7)	34,597 (76.5)	4771	9892	24,730	49,689	82,893	1398 (42.1)	2790 (35.8)	750	1340	3500	6340	10,542	1510 (43.6)	12,517 (64.9)	507	957	2331	14,711	36,119
Outpatient procedures	573 (15.0)	5967 (13.2)	1355	2443	4593	7617	11,726	576 (17.4)	1988 (25.5)	19	547	1309	2347	4070	741 (21.4)	3403 (17.6)	750	1455	2461	4163	7087
Hospice	83 (2.2)	268 (0.6)	0	0	0	0	0	111 (3.3)	237 (3.0)	0	0	0	0	0	59 (1.7)	200 (1.0)	0	0	0	0	0
Total cost	3814 (100.0)	45,247 (100.0)	8707	16,657	37,878	60,276	91,313	3318 (100.0)	7804 (100.0)	2939	5254	7641	13,283	17,852	3464 (100.0)	19,301 (100.0)	2846	5067	10,249	24,317	47,118

**Table 4 cancers-16-00592-t004:** Costs related to the overall disease follow-up (from the start of 1L treatment until the end, last visit, or death) were reported in PPPM (€) and other descriptive statistics, grouped by mutation status. The negative/unknown cohort was further sub-grouped based on 1L starting time in pre- and post-1L IO.

Overall Costs	EGFR, ALK, or ROS1 Mutation-Positive Patients							EGFR, ALK, or ROS1 Negative/Unknown Patients Pre-1L IO							EGFR, ALK, or ROS1 Negative/Unknown Patients Post-1L IO						
	Mean PPPM € (%)	Mean Cost € (%)	Percentiles of the Costs Distribution					Mean PPPM € (%)	Mean Cost € (%)	Percentiles of the Costs Distribution					Mean PPPM € (%)	Mean Cost € (%)	Percentiles of the Costs Distribution				
			10%	25%	50%	75%	90%			10%	25%	50%	75%	90%			10%	25%	50%	75%	90%
Hospitalization	631 (17.2)	7290 (10.2)	0	0	4508	12,116	17,241	1376 (45.2)	6926 (35.2)	0	1758	4161	9230	16,204	1179 (36.4)	6413 (22.1)	0	0	4161	9129	17,863
Cancer therapy	2414 (65.7)	53,895 (75.9)	6739	15,588	35,140	62,606	132,707	914 (30.0)	7187 (36.6)	800	1760	4186	8877	14,216	1333 (41.2)	16,424 (56.4)	546	1445	6204	21,917	44,745
Outpatient procedures	484 (13.2)	8556 (12.1)	2276	4196	7256	11,123	16,905	458 (15.0)	3834 (19.5)	122	778	2125	4533	8788	571 (17.6)	5475 (18.8)	1147	2010	3752	7753	12,065
Hospice	144 (3.9)	1244 (1.8)	0	0	0	394	5516	298 (9.8)	1702 (8.7)	0	0	0	1576	4985	155 (4.8)	799 (2.7)	0	0	0	197	2659
Total cost	3673 (100.0)	70,985 (100.0)	15,311	33,562	56,055	81,266	152,708	3046 (100.0)	19,649 (100.0)	5505	8721	14,815	23,794	35,389	3238 (100.0)	29,111 (100.0)	5408	10,572	20,978	38,490	60,314

**Table 5 cancers-16-00592-t005:** Healthcare resources utilization (HCRU) related to the 1L line of treatment was reported as PPPM (€) and other descriptive statistics, grouped by mutation status. The negative/unknown cohort was further sub-grouped based on 1L starting time in pre- and post-1L IO.

1L HCRU	EGFR, ALK, or ROS1 Mutation-Positive Patients							EGFR, ALK, or ROS1 Negative/Unknown Patients Pre-1L IO							EGFR, ALK, or ROS1 Negative/Unknown Patients Post-1L IO						
	PPPM	Mean HCRU	Percentiles of HCRU Distribution					PPPM	Mean HCRU	Percentiles of HCRU Distribution					PPPM	Mean HCRU	Percentiles of HCRU Distribution				
			10%	25%	50%	75%	90%			10%	25%	50%	75%	90%			10%	25%	50%	75%	90%
Number of Hospitalization	0.1	1.0	0.0	0.0	1.0	1.0	3.0	0.27	0.6	0.0	0	0.0	1.0	2.0	0.3	0.8	0.0	0.0	0.0	1.0	2.0
Hospitalization LOS	1.0	6.9	0.0	0.0	2.0	10.0	21.0	3.24	6.7	0.0	0.0	0.0	9.0	20.0	3.1	8.0	0.0	0.0	0.0	13.0	22.5
Number of cancer therapy administrations	1.9	17.4	0.0 *	3.0	15.0	28.0	37.0	4.75	16.20	1.0	1.6	12.0	23.0	34.0	2.5	13.6	3.0	6.0	9.0	17.0	28.0
Number of outpatients visit	2.4	27.2	6.0	10.0	20.0	38.0	55.0	2.33	8.2	1.0	2.0	5.0	11.0	18.0	2.9	13.7	4.0	5.0	9.0	16.0	27.5
Hospice LOS	0.4	1.3	0.0	0.0	0.0	0.0	0.0	0.53	1.2	0.0	0.0	0.0	0.0	0.0	0.3	1.0	0.0	0.0	0	0.0	0.0
ER admissions	0.2	1.2	0.0	0.0	1.0	2.0	4.0	0.37	1.1	0.0	0.0	1.0	2.0	3.0	0.4	1.0	0.0	0.0	1.0	2.0	3.0

* HCRUs associated with 1L cancer therapy administrations were not detected for 14 mutation-positive patients treated during the abolition of day-hospital stays (April 2016).

**Table 6 cancers-16-00592-t006:** Healthcare resources utilization (HCRU) related to the overall disease follow-up (from the start of 1L treatment until the end, last visit, or death) was reported as PPPM (€) and other descriptive statistics, grouped by mutation status. The negative/unknown cohort was further sub-grouped based on 1L starting time in pre- and post-1L IO.

Overall HCRU	EGFR, ALK, or ROS1 Mutation-Positive Patients							EGFR, ALK, or ROS1 Negative/Unknown Patients Pre-1L IO							EGFR, ALK, or ROS1 Negative/Unknown Patients Post-1L IO						
	PPPM	Mean HCRU	Percentiles of HCRU Distribution					PPPM	Mean HCRU	Percentiles of HCRU Distribution					PPPM	Mean HCRU	Percentiles of HCRU Distribution				
			10%	25%	50%	75%	90%			10%	25%	50%	75%	90%			10%	25%	50%	75%	90%
Number of Hospitalization	0.2	1.7	0.0	0.0	1.0	3.0	4.0	0.33	1.7	0.0	1.0	1.0	2.0	4.0	0.3	1.6	0.0	0.0	1.0	2.0	4.0
Hospitalization LOS	1.2	14.0	0.0	0.0	9.0	22.0	36.0	3.34	14.9	0.0	2.0	10.0	21.0	33.0	3.1	15.5	0.0	0.0	9.0	23.0	39.0
Number of cancer therapy administrations	1.4	28.4	2.0	10.0	24.0	37.0	68.0	2.94	20.5	1.0	6.0	15.0	26.0	41.0	2.1	21.4	4.0	7.0	15.0	29.0	48.5
Number of outpatients visit	2.2	40.5	10.0	18.0	32.0	54.0	90.0	2.10	18.8	1.0	3.0	10.0	23.0	44.0	2.4	22.9	4.0	9.0	15.5	32.0	49.5
Hospice LOS	0.7	6.1	0.0	0.0	0.0	2.0	27.0	1.48	8.5	0.0	0.0	0.0	8.0	25.0	0.8	4.0	0.0	0.0	0.0	2.0	13.0
ER admissions	0.1	1.9	0.0	0.0	1.0	3.0	5.0	0.33	2.0	0.0	1.0	1.0	3.0	4.0	0.3	1.6	0.0	0.0	1.0	2.0	3.5

## Data Availability

The raw data presented in this study are available at a reasonable request from the corresponding author.

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
