# Peer review of "Healthcare Costs and Resource Utilisation of Italian Metastatic Non-Small Cell Lung Cancer Patients"

_cancers, 2024, doi:10.3390/cancers16030592_

Round 1

Reviewer 1 Report

Comments and Suggestions for Authors

The analysis of costs to be addressed in treatment pathways for diseases with an inauspicious outcome is an ever-present topic with important implications regarding cost-effectiveness, benefit, and utility.

This article appears well structured: the introduction provides sufficient elements for an overview of the topic. The methods appear adequate, and the writing of the results is consistent and easy to read.

I found no issues to highlight. I congratulate the authors for their work. 

Author Response

The authors thank the reviewer, for this positive feedback and for taking the time to evaluate the manuscript.

Reviewer 2 Report

Comments and Suggestions for Authors

Dear authors

I would like to thank you for giving me the opportunity to review the manuscript entitled “Healthcare costs and resources utilisation of Italian metastatic non-small cell lung cancer patients”. The study, evaluating the economic burden of metastatic non-small cell lung cancer patients in the context of immuno-oncology regimens, provides valuable insights into the healthcare costs and resource utilization dynamics. I believe the findings will make a significant contribution to our understanding of this critical aspect of cancer care.

Best regards

Author Response

The authors thank the reviewer for this positive feedback and for taking the time to evaluate the manuscript.

Reviewer 3 Report

Comments and Suggestions for Authors

This is a valuable study, however, the message is not sufficiently clearly stated in the current version of the manuscript. It is self-explanatory that the incorporation of immunooncology drugs, e.g., pembrolizumab, increases both expenses and survival. The major question if this increase is more or less proportional, i.e., a proper quantitative comparison of both these increments is highly essential. This issue needs to be addressed with the utmost clarity, with Simple Summary and the Abstract giving the answers.

The same applies to patients with oncogenic drivers. I suggest to change their naming from “positive” to “mutation-positive” of “mutation/fusion-positive”. Obviously, they almost always respond to the tablets, and the response is long, so there is no wonder that expenses are high. Again, is there proportional increase with respect to the survival, or not? What are the results if one compares 3rd generation EGFR inhibitors given in the 1st line versus older EGFR antagonists?

Table 2: what the meaning of the arrow sign?   

Comments on the Quality of English Language

Minor editing required

Author Response

We thank the reviewer for this relevant comment, which allowed us to further reflect on the results obtained. In fact, during the months of the analysis, we had already hypothesized a proper quantitative comparison as cost-effectiveness analysis of the introduction of immuno-oncology drugs. However, it would have been impossible to do it properly without introducing some bias. Specifically, it would have been optimal to carry out a comparison between the cohort of patients treated with a first-line PD-1/PD-L1 inhibitor and a population eligible for the same treatment in the period prior to the approval of PD-1/PD -L1 inhibitor in the first line (i.e. a subset of patients among the 229 patients of the “Pre-1L IO” cohort). Nevertheless, the availability of data on the mutational status of PD-1/PD is a fundamental requirement to select these patients (and then carry out a propensity score matching to balance the groups). Unfortunately, PD-1/PD-L1 determination only became routine when PD-1/PD-L1 inhibitor became available in the first line.

A possible alternative would have been the comparison between patients treated in first line with PD-1/PD-L1 inhibitor (i.e. the 67 patients treated with single agent PD-1/PD-L1 inhibitor plus the 19 patients treated in first line with PD-1/PDL1 inhibitor + chemotherapy) and those among the “Pre-1L IO” cohort (EGFR, ALK, or ROS1 negative) who received a PD-1/PDL1 inhibitor drug in subsequent lines, thus ensuring PDL1 positivity (i.e. >1). However, this approach would have introduced a substantial immortal time bias. Furthermore, this immortal time bias would have been introduced only in one of the two groups compared.

Finally, the last hypothesized methodological approach was to compare between the entire cohort of 290 "Post-1L IO" patients and the entire cohort of 229 "Pre-1L IO" patients. It is necessary to premise that a correct economic evaluation (for example a cost-effectiveness analysis) cannot ignore the identification of the target population towards which the intervention or health technology (e.g. drug, screening, device) is aimed. In both cohorts, we would have included patients with negative PDL1 (i.e. 0 and <1) and therefore not treatable with the drugs of interest. Furthermore, while in the "Post-1L IO" population we would have been able to identify the eligible population thanks to the PDL1 data, this would not have been possible in the "Pre-1L IO" population, making the two populations not comparable effectively and potentially introducing a selection bias. Lastly, even assuming a balance between the two groups, due to the population not being entirely eligible for treatment with a PD-1/PDL1 inhibitor, the effect would have been biased (overestimated or underestimated). 

  • As regards the comparison between the 3rd generation EGFR inhibitors given in the 1st line versus older EGFR antagonists, a comparison between patients treated with osimertinib and those treated with older EGFR antagonists could have been performed. However, in our population only 19 patients received 3rd generation EGFR inhibitors in the 1st line and the sample size would not have allowed an economic evaluation with reliable results, even by performing a sensitivity analysis. However, we considered the proposal very interesting, and it can be developed in future research when the population treated with 3rd generation EGFR inhibitors in 1st line in our institute becomes numerically sufficient. As suggested by the reviewer, we changed the name of the "positive" group to " mutation-positive" in all the manuscript.
  • About Table 2, the arrow signs were a misprint. We replaced the arrow symbol with the high dash.

Reviewer 4 Report

Comments and Suggestions for Authors

The paper is very well done. 

This paper examines the economic costs of metastatic non-small cell lung cancer on Italian patients before and after IO regimens as the first line of treatment. 

Line 54-55 should be rewritten, it is a bit repetitive

Why didn't this study incorporate a Cost-effectiveness or Cost-utility component? All the data is there for this but it is not presented. Please explain or incorporate into the paper.

The conclusions section is not sufficient. Of course NSCLC is expensive, a study isn't needed for this. Reinforce further what your results show and why it is important. This section is really lacking

Author Response

We thank the reviewer for his/her valuable comments. These comments are very constructive and helped us to improve the manuscript. 

  1. We re-write the lines 54-55 as suggested;
  2. We thank the reviewer for this relevant comment, which allowed us to further reflect on the results obtained. In fact, during the months of the analysis, we had already hypothesized a cost-effectiveness analysis of the introduction of immuno-oncology drugs. However, it would have been impossible to do it properly without introducing some bias. Specifically, it would have been optimal to carry out a comparison between the cohort of patients treated with a first-line PD-1/PD-L1 inhibitor and a population eligible for the same treatment in the period prior to the approval of PD-1/PD -L1 inhibitor in the first line (i.e. a subset of patients among the 229 patients of the “Pre-1L IO” cohort). Nevertheless, the availability of data on the mutational status of PD-1/PD is a fundamental requirement to select these patients (and then carry out a propensity score matching to balance the groups). Unfortunately, PD-1/PD-L1 determination only became routine when PD-1/PD-L1 inhibitor became available in the first line.

    A possible alternative would have been the comparison between patients treated in first line with PD-1/PD-L1 inhibitor (i.e. the 67 patients treated with single agent PD-1/PD-L1 inhibitor plus the 19 patients treated in first line with PD-1/PDL1 inhibitor + chemotherapy) and those among the “Pre-1L IO” cohort (EGFR, ALK, or ROS1 negative) who received a PD-1/PDL1 inhibitor drug in subsequent lines, thus ensuring PDL1 positivity (i.e. >1%). However, this approach would have introduced a substantial immortal time bias. Furthermore, this immortal time bias would have been introduced only in one of the two groups compared.

    Finally, the last hypothesized methodological approach was to compare between the entire cohort of 290 "Post-1L IO" patients and the entire cohort of 229 "Pre-1L IO" patients. It is necessary to premise that a correct economic evaluation (for example a cost-effectiveness analysis) cannot ignore the identification of the target population towards which the intervention or health technology (e.g. drug, screening, device) is aimed. In both cohorts, we would have included patients with negative PDL1 (i.e. 0%) and therefore not treatable with the drugs of interest. Furthermore, while in the "Post-1L IO" population we would have been able to identify the eligible population thanks to the PDL1 data, this would not have been possible in the "Pre-1L IO" population, making the two populations not comparable effectively and potentially introducing a selection bias. Lastly, even assuming a balance between the two groups, due to the population not being entirely eligible for treatment with a PD-1/PDL1 inhibitor, the effect would have been biased (overestimated or underestimated). Unfortunately, it is impossible to perform a cost utility analysis, since data on quality of life to determine QALYs are not available.
  3. We addressed the reviewer's concern about the conclusion section, revising the conclusion to include additional details as suggested.

Round 2

Reviewer 3 Report

Comments and Suggestions for Authors

The authors have considered the suggestions

Comments on the Quality of English Language

Minor editing required